# A Compact Sub-GHz Wide Tunable Antenna Design for IoT Applications

**Rifaqat Hussain [1]**, **Saad I. Alhuwaimel [2]**, **Abdullah M. Algarni [1]**, **Khaled Aljaloud [3]** and **Niamat Hussain [4,*]**

1   Electrical Engineering Department, King Fahd University for Petroleum and Minerals (KFUPM), Dhahran 31261, Saudi Arabia; rifaqat@kfupm.edu.sa (R.H.); algarnia@kfupm.edu.sa (A.M.A.)
2   King Abdulaziz City for Science and Technology, Riyadh 12354, Saudi Arabia; huwaimel@kacst.edu.sa
3   College of Engineering, Muzahimiyah Branch, King Saud University, P.O. Box 2454, Riyadh 11451, Saudi Arabia; kaljaloud@ksu.edu.sa
4   Department of Smart Device Engineering, School of Intelligent Mechatronics Engineering, Sejong University, Seoul 05006, Korea
*   Correspondence: niamathussain@sejong.ac.kr

**Abstract:** This work presents a compact meandered loop slot-line 5G antenna for Internet of Things (IoT) applications. Recently, sub-gigahertz (sub-GHz) IoT technology is widely spreading. It enables long-range communications with low power consumption. The proposed antenna structure is optimized to operate at sub-GHz bands without any additional complex biasing circuitry or antenna structure. A miniaturized design was achieved by a meandered structured loop slot-line that is loaded reactively with a varactor diode. Wideband frequency reconfigurability (FR) was achieved by the use of the varactor diode. The proposed antenna resonates over the frequency band of 758–1034 MHz with a minimum bandwidth of 17 MHz over the entire frequency band. The RO4350 substrate with dimensions of $0.18\lambda_g \times 0.13\lambda_g$ mm$^2$ is used to design the proposed antenna design. The efficiency and gain values varied from 54–67% and 0.86–1.8 dBi. Compact planar structure, narrow-band operation (suitable for NB-IoT) and simple biasing circuitry, which allows for sub-GHz operation, are unique and attractive features of the design.

**Keywords:** sub-GHz; meandered loop; frequency agile; IoT antenna; 5G antenna

## 1. Introduction

The exponential Internet of things (IoT) growth with the use of small-cell devices results in the massive expansion of the traffic loads. This demand has revolutionized the current technology into 5G, which requires ultra-low latency, a high data rate, and improved capacity. By 2030, multibillion IoT technology users are expected. The growth of sub-GHz low-cost and reliable RF front-end devices is increasing as a requirement of this massive connectivity. Such systems are good candidates as the spectrum below 1 GHz is less crowded and reliable propagation can be ensured in large infrastructures. The Global System for Mobile Communications (GSMA) industry completely described the details of 3GPP standards for both NB-IoT and LTE-M technologies as part of the developing specifications for 5G. NB-IoT and LTE-M technologies are continuously evolving as a part of 5G as per 3GPP. Both technologies are suitable for 5G-enabled NB-IoT devices.

NB-IoT applications include street lighting, electronic appliances, sensors, actuators, and other machines that can be connected to the internet and with each other through device-to-device (D2D) connectivity. NB-IoT networks are suitable for a large number of connected devices with extended battery life. For such applications, long-range and low-power consumption communications are required to work in the sub-GHz bands. As a result, designing compact structure sub-GHz antenna systems that work over multiple frequency bands lower than 1 GHz has become a necessity. This is attainable using frequency reconfigurable (FR) antennas that have omni-directional radiation characteristics, which

extend the antenna coverage area [1]. The proposed NB-IoT antenna design is suitable for numerous applications, such as healthcare, pet tracking, kid monitoring, smart metering, parking alarms, event detectors, and home appliances.

With the fast-growing IoT applications, the need for efficient power management and communication over long distances are challenging requirements. In the literature, several IoT antenna designs that address various IoT application challenges at different frequencies have been reported. The reported solutions include compact and low-profile antenna designs for multi-standard IoT antennas with wide-range frequency coverage in sub-6 GHz bands. These various antenna designs include inverted-F antennas, loop antennas, and monopole antennas.

Antenna designs based on monopoles for IoT applications were presented in [2–4]. Such solutions could be appropriate for wideband antenna operation intended to cover many wireless communication systems standards. In [2], a compact (20 mm × 30 mm) multi-band single element antenna is presented. The proposed antenna covers the following frequency bands: 1.79–2.63 GHz, 3.46–3.97 GHz, 4.92–5.85 GHz, and 7.87–8.40 GHz. This is an integrated 4G/5G antenna design to operate at sub-6 GHz and in the mm-wave band. Slots and monopoles were utilized to achieve the desired integrated operation. This is one of the pioneering works in 4G/5G integrated solutions. A compact structure monopole antenna that is resonating over the frequency band from 3 to 12 GHz, the board dimensions are 9.45 mm × 18.5 mm was presented in [3]. In this work, a slot antenna based 4G/5G integrated solution was provided using a dual-function planar connected array. The authors in [4], a multi-band antenna was presented for IoT applications. The main antenna features were its suitability for near-field communications in both microwave frequency bands and ultra-high frequency (UHF) bands. In addition, the antenna is suitable for low-frequency bands operating in IoT applications. This work also reported a 4G/5G integrated solution based on a monopole antenna array.

Several patch antenna designs for IoT applications were investigated in [5,6]. A 2.4 GHz high fractional bandwidth compact patch antenna was presented in [5]. The proposed antenna consists of an inverse S-shaped meander line that is connected to the rectangular box with a slot. This technique is utilized to reduce the antenna dimensions. The board dimensions are 40 mm × 1.6 mm. Both parasitic patch and capacitive loading were used to achieve a gain of −0.256 dBi and an overall efficiency of 79%. The work in [6] presented a miniaturized patch antenna design that operates in sub-GHz. A 95% miniaturization was achieved, compared to a conventional patch antenna, over the frequency band 805 to 835 MHz. The use of high folding, slots, and slits with inductive loading, utilizing vias, helped in achieving this excellent miniaturization. The dimensions of the antenna board with the elevated structure are 30 mm × 31.84 mm × 4.37 mm.

Other antenna types for IoT applications are reported in the literature. For example, the following: monopole antennas loaded with inverted-L shaped stubs in [7], 3D-printed antennas [8], loop antennas [9], and glass frame antennas [10]. The solutions provided in [7,8] are based on monopole antennas and non-planar antenna structures. In [9], a 2.45 GHz transparent conductor antenna for IoT applications was presented. The variations in the realized antenna gain in the presence of metal frames nearby have been discussed in detail. The antenna is compact in size, with dimensions of 36 mm × 36 mm × 0.5 mm. In this work, a dual-band shared-aperture based antenna is presented with high channel isolation. Some other recent work on antenna designs for IoT applications was also reported in [11–14]. All these antennas operate above 1 GHz and hence are not suitable for narrow band IoT applications for long-range communication.

Most of the antenna designs operating in sub-GHz are elevated printed inverted F-antennas (PIFA), monopole, or dipole antennas [15–19]. In [15], a folded miniaturized patch antenna was presented for IoT device operating at 805–835 MHz. In [16], digitally tunable capacitor (DTC) was used to tune the antenna from 600–900 MHz. The digital control and the low power consumption make this antenna a good candidate for IoT applications. In [17], a transparent double folded loop antenna was presented for IoT applications. The

antenna was operating at 2.4 GHz with a bandwidth of 500 MHz. In [18], a 3D printed antenna-on-package antenna was presented with isotropic patterns for an IoT application operating at 2.4 GHz. Similarly, in [19], an energy harvesting rectenna was presented for IoT applications. The proposed antenna design would utilize an energy harvesting mechanism using rectenna to mitigate the challenges of battery constraint issues. The antenna was operated at a frequency band of 2.4 GHz.

Also, other frequency reconfigurable antennas closely resembling the proposed antenna are reported in [20–23]. The solutions provided are based on either reactive loading or meandering structures. However, neither of them was able to achieve a compact planar structure with sub-GHz operation. Also, the solutions provided in [20–23] are non-planar structures or non-reconfigurable structures with large antenna dimensions.

Thus, the non-compact and non-planar antennae limited the tuning capabilities, affecting their suitability for small IoT applications and terminal devices. Only a few slot-based antenna designs that operate in sub-GHz bands are reported in the literature, such as [24,25]. The solutions provided in [24,25] used a simple slot structure to tune the antenna in the desired frequency. Most of the reported frequency reconfigurable antenna designs are operating in frequency bands that are above 2 GHz. The proposed design in [24] covers the frequency bands of 0.9, 1.8, 1.9, and 2.4 GHz. The antenna size is 38 mm $\times$ 16 mm. In [25], a dual antenna design that covers the 0.860 and 2.45 GHz frequency bands was presented. The antenna is 50 mm $\times$ 50 mm in size.

In [26], an energy harvesting antenna design for IoT applications was presented. The proposed antenna consisted of a rectangular patch along with a filter and rectifier circuitry operating at a frequency band of 2.4 GHz. The authors provided an energy harvesting solution without optimizing the antenna. The authors in [27] present a 60 GHz mm-wave antenna for the IoT. The total bandwidth of this antenna is 9.8 GHz having a peak gain of 9.6 dBi. The performance of the antenna was thoroughly evaluated to determine the radiation coverage. The given antenna design is a well-suited option for body-centric IoT applications because of its compact structure.

Very few antennas for sub-GHz were reported, as it is quite challenging to design a compact antenna structure. This is because of the poor input impedance matching with the small antenna size. In [28], a detailed survey based on fractal antennas was presented. The significance of the fractal antennas for IoT application is well studies and provided a thorough literature survey on fractal antennas. Various IoT antennas are reported in the literature [29–31]. In [29], a transparent 2-element 5G MIMO antenna for sub-6 GHz applications is described. Thus, this work provided a guideline on the use of transparent for IoT applications. Similarly, a low-profile single-band and dual-band antenna were presented in [30] for IoT applications. The slot structure was optimized to operation in single as well as dual band. in [31], the authors proposed an artificial magnetic conductor-backed compact wearable antenna was proposed. Such antennas are good candidates for IoT applications. At the expense of complex antenna geometry. Comparing to other works, the proposed antenna design outperformed in terms of various features are comprehensively discussed in this article.

The novelty and distinguishing features of the proposed work are described in detail as follows:

(1) To overcome the challenging requirement for IoT devices to have extended coverage with minimal power consumption, it is highly desirable to design a compact antenna for lower frequency bands. The proposed antenna structure is an attempt to meet the desired performance and standards of narrow-band IoT (NB-IoT) applications for 5G-enabled IoT devices. None of the available literature [2–6,15–23], and [32–36] can be used for NB-IoT in sub-GHz with wide tuning capability;

(2) A compact meandered loop slot-line antenna with frequency agility suitable for NB-IoT applications that operate in sub-GHz bands is proposed. The work cited in [4,6,15–19,21,23] covered sub-GHz bands, but the majority of them were not suitable for NB-IoT operations;

(3)　The proposed antenna benefits from the following unique features: simple biasing circuitry, compact planar structure, and it operates over the sub-GHz bands which fit for NB operations, but most of the existing antennas are not suitable for NB operation [2,3,6,16–19,21–23,32–36];

(4)　Miniaturization was achieved using a unique combination of meandered loop slot-line along with a reactively loaded slot antenna. A 173% size reduction in area is obtained using this technique. As per the authors' information, none of the works achieved such a high level of miniaturization;

(5)　Resonating bands' smooth variation is noticed over a very wide band from 758 to 1034 MHz with antenna dimensions of $60 \times 27$ mm$^2$;

(6)　Moreover, a step-by-step antenna design procedure and analysis are provided to give general guidelines to understanding how to scale its design for any desired frequency band;

(7)　As per 3GPP Release 13 and 15, the proposed antenna design is suitable to be utilized in the following NB-IoT bands: B-5, B-8, B-13, B-14, B-17, B-18, B-19, B-20, and B-26. None of the reported literature can cover such a large number of sub-1 GHz bands.

(8)　The proposed NB-IoT antenna is best suited for 5G-enabled NB-IoT devices.

Section 2 of this paper discusses the proposed antenna design details and the theoretical analysis. The simulated and measured scattering parameters results and the radiation characteristics of the proposed antenna design are presented in Section 3. Finally, Section 4 provides a set of conclusions about the presented work.

## 2. Design Details

The major challenges for IoT antenna designs are the requirements for compact antenna structures and sub-GHz operations for better power management along with long-distance communication. In this section, the detailed antenna geometry of the proposed design is presented, equivalent circuit diagram and the physics behind the antenna's operation, followed by a step-by-step antenna design procedure.

### 2.1. Antenna Geometry

Figure 1 shows the proposed a meandered loop slot-line antenna design. The antenna was designed and fabricated using an RO4350 substrate with dimensions of 27 mm $\times$ 60 mm $\times$ 0.76 mm. The relative permittivity ($\varepsilon_r$) of the substrate is 3.48 and its loss tangent (tan$\delta$) is 0.0036. The Rogers substrate can be replaced with FR-4 to obtain good resonance sweeps but with slightly lower radiation efficiency values. For the Rogers-4350 board, the temperature coefficient of the dielectric constant is among the lowest of any circuit board material. This makes it an ideal substrate for broadband applications. After optimizing the antenna design, the design is mainly composed of the following: an etched slot-line structure from the ground plane and a meandered-shaped rectangular slot-line used to increase the radiating structure's effective electrical length. The proposed antenna design dimensions are the following: $l_1 = 38$, $l_2 = 36$, $l_3 = 19$, $l_4 = 11$, $l_5 = 7.5$, $w_1 = 5.5$, $w_2 = 3.5$, $w_3 = 6$, $w_4 = 4$.

The proposed antenna top layer detailed view is shown in Figure 1a. It is composed of a matched 50 $\Omega$ input impedance ($Z_{in}$), a microstrip feedline, and the biasing circuitry of the varactor diode. The varactor diode biasing circuit ($Dv$) consisted of a series of combinations of RF chokes ($L_1$ and $L_2$) and current limiting resistors ($R_1$ and $R_2$). Sorting posts (SP) were used to connect the bottom layer varactor diode to the biasing circuitry in the top layer. RF chokes were used to isolate the radiating structure from the power supply. The reverse-biased varactor diode acted as a DC blocking capacitor. Hence, the DC biasing part and the RF radiating structure are well isolated and have a very minimal effect on the antenna's performance. SMV 1233 varactor diode is used [37]. The varactor diode (SMV1233) maximum reverse bias current is 20 nA. From the maximum current specifications, the maximum biasing circuit power loss is $-87.7$ dBm.

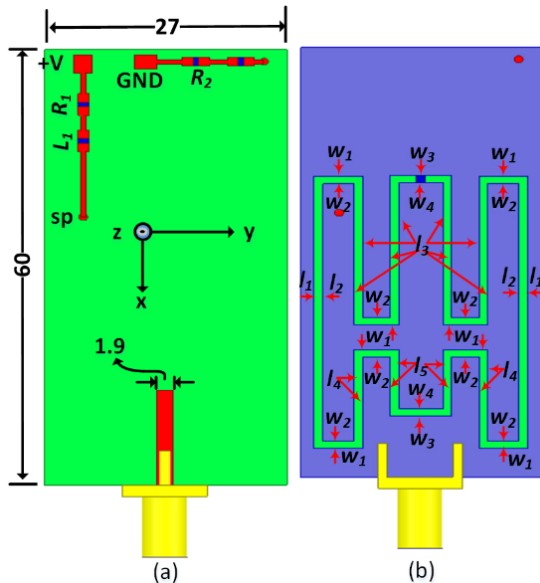

**Figure 1.** Proposed antenna (**a**) Top view (**b**) Bottom view (All dimensions are in millimeters, mm).

The ground plane detailed view is shown in Figure 1b. After optimization, the rectangular slot operates at below 1 GHz frequency band. After several parametric sweeps of the structure's length and width, the resultant structure is shown. The 1 mm wide uniform slot-line is used throughout the meandered structure. The final optimized antenna dimensions are shown in Figure 1b. The varactor diodes' exact positions and placements are critical as they help in achieving wide frequency band tuning. The LPKF S103 machine [38] was used to fabricate the antenna. The fabricated model top and bottom layers are shown in Figure 2a,b, respectively. The other antenna design dimensions are shown in Figure 1. An increase in the slot length will increase the electrical length of the antenna, and hence the antenna will resonate at a lower frequency band. On the other hand, any change in the slot width will result in poor matching. The given optimal slot width gave us the best $Z_{in}$ matching.

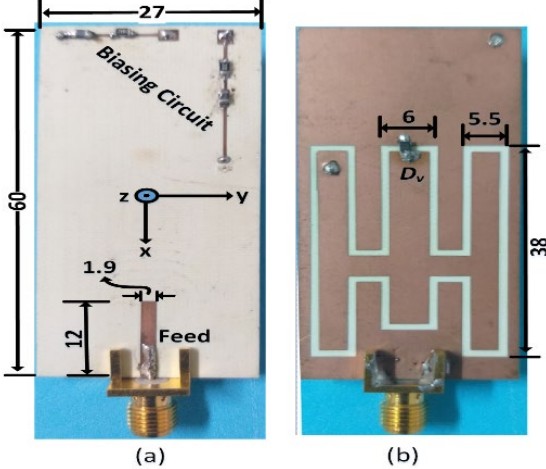

**Figure 2.** Fabricated antenna (**a**) Top view (**b**) Bottom view (dimensions are in millimeters, mm).

## 2.2. Antenna Operation

Slot-based FR antennas gain more attraction as they are low-profile planar structures, easy to integrate, and able to operate in wide frequency bands. In the literature, open and closed-ended slot designs were investigated and reported. The closed ends or short-circuited slot antenna can be modeled as $\lambda/2$ transmission line which corresponds to its fundamental frequency of resonance [39]. These antennas may be loaded with capacitive

reactance. The antenna resonance can be optimized over a wide frequency band. The rectangular slot antenna fundamental resonance frequency is given by the following [40]:

$$f_r = \frac{c}{(l_m + w_m)} \times \frac{c}{4 \, f_{res} \, \sqrt{\frac{\varepsilon_r + 1}{1.5\varepsilon_r}}} \qquad (1)$$

$l_m = l_1 + l_2 + 4l_3 + 2l_4 + 2l_5$, $w_m = 4w_1 + 4w_2 + w_3 + w_4$, $c$ is the speed of light in free space, $\varepsilon r$ is the relative permittivity of the substrate, $f_r$ is the modified rectangular antenna's fundamental resonance frequency, and the term $(l_m + w_m)$ is the mean circumference of the meandered rectangular ring slot antenna. In the proposed design, the effective mean circumference can be an estimate of several rectangular slots that determine the overall fundamental resonance frequency.

The meandered loop-slot structure was reactively loaded using a varactor diode $(D_v)$, as shown in Figures 1 and 2. The varactor diode was utilized to vary the slot-line capacitance at a certain point. As a result, it helped in bringing the resonance frequency down to the lower band. The reactive loading is a non-uniform operation, and it can be determined using the location of the varactor diode $(L_{v1})$, its capacitance value $C_v$ and the impedance $(Z_o)$ of the slot line structure. The resonance frequency of the reactive loaded slot can be determined using the transmission line equivalent circuit model as given in the following [41]:

$$\tan(\beta \, L_t) + \tan(\beta \, (L_t - L_v)) - \omega C_v Z_{in} \tan(\beta \, L_1) \tan\beta \, (L_t - L_v) = 0 \qquad (2)$$

where $L_t$ is the total length of the slot $(L_t = l_m + w_m)$, $\beta$ is the propagation constant. This constant depends on the frequency of operation $L_v$ is the distance of diode placement from the feedline, $C_v$ is the varactor diode capacitance, and $\omega$ is the angular frequency. The reactively loaded slot antenna resonance frequency can be numerically determined by solving Equation (2). The resonance frequencies obtained for the optimized design using HFSS were also compared with theoretical values based on Equation (2). The two results (theoretical, HFSS-based) obtained are (750 MHz and 764 MHz), (826 MHz, 836 MHz), (901 MHz, 885 MHz), (981 MHz, 964 MHz), (1052 MHz, 1029 MHz). A small variation in values is observed. Thus, the mathematical expression derived is useful to obtain an insight into the effective reactive loading of the slot antenna structure.

Figure 3 shows the equivalent circuit model of the proposed meandered loop-slot antenna. The series combination of a microstrip feedline (series $L_f C_f$ circuit) and an *RLC* resonating circuit that represents the loop-slot radiating structure is shown in Figure 3a.

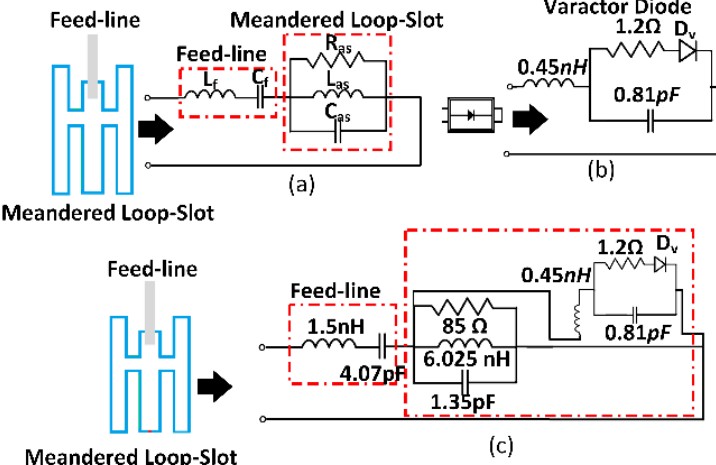

**Figure 3.** Equivalent circuit model (**a**) Meandered loop-slot with feedline (**b**) Varactor diode (**c**) Complete circuit model.

The varactor diode equivalent circuit and the complete antenna design equivalent circuit model are shown in Figure 3b,c, respectively [42]. The antenna equivalent circuit model analysis assists in gaining a better understanding of the proposed antenna design multi-band operation, the slot reactive loading, and FR antenna operation. ADS can be used to obtain the antenna circuit element values and the antenna reactive loading parts can be extracted by using the S-parameters [43].

### 2.3. Antenna Design Procedure

This section provides a step-by-step antenna design procedure and optimization steps to obtain the final design. The first step to designing the proposed antenna was designing a rectangular slot-line antenna structure with dimensions 24 mm × 15 mm, fed with 50 Ω microstrip line. The antenna was resonating above the 2 GHz bands. The dimensions of the slot were optimized to make it resonate above 1.5 GHz by increasing the electrical length of the radiating slot. To further reduce the resonance frequency to lower bands, the rectangular structure was turned into a meandered structure to achieve a resonance frequency that is below 2 GHz, as shown in Figure 4a. For the given meandered structure, the slot structure was resonating at 1.75 GHz. The antenna was further optimized by increasing its electrical length, as shown in Figure 4b. The given structure was working at 1.305 GHz. The width of each slot as well as the distance between different meandered slots were optimized to tune the antenna to be effectively loaded with varactor capacitance to cover the maximum sub-GHz bands of the proposed antenna structure. For the same antenna structure as shown in Figure 4b, the theoretical resonance value calculated based on Equation 1 was 1.287 GHz. Hence, a very close agreement between the theoretical and simulated values was observed.

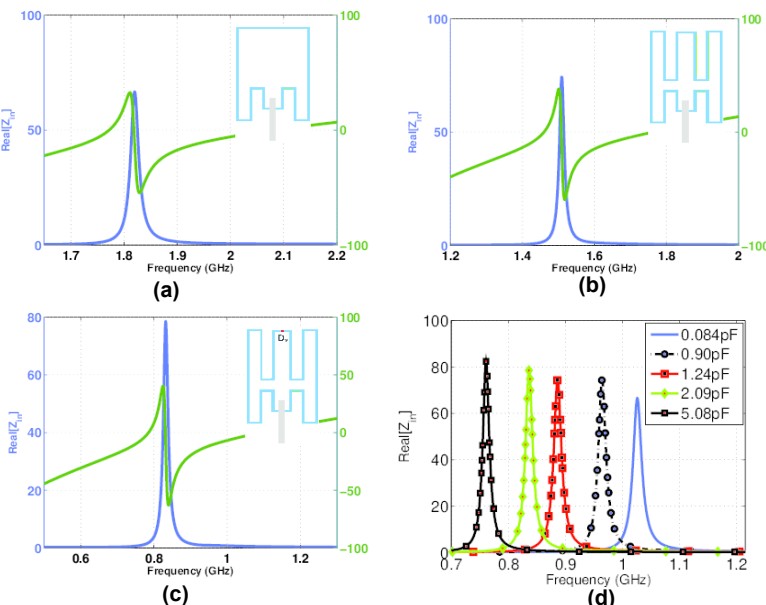

**Figure 4.** Zin curves (**a**) Stage-1 $Z_{in}$ curves (**b**) $Z_{in}$ curves of the optimized design (**c**) $Z_{in}$ curves with varactor loading (**d**) Re{$Z_{in}$} for various capacitance values.

A single varactor diode was utilized to bring the resonance frequency down to sub-GHz bands, as well as a continuous sweep of frequencies was obtained over a wide band. Figure 4a–c show the real($R_e$) and imaginary($I_m$) parts of input impedance ($Z_{in}$). It can be seen that Re{$Z_{in}$} is around 50Ω while the Im{$Z_{in}$} is crossing the zero value at the resonating bands. The placement of the diode was very critical as the input impedance ($Z_{in}$) matching was dependent on it. The varactor diode's various locations resulted in different reactive loading effects on the radiating structure. To obtain an optimum Zin matching, several parametric analyses were performed, including its placement on the slot structure.

The antenna optimized dimensions and the placement of diodes resulted in a continuous frequency sweep from 758~1034 MHz, which can thus support several narrow-band IoT applications below 1 GHz. Figure 4d shows the Re{$Z_{in}$} for different capacitance values of 0.84, 0.90, 1.24, 2.09, and 5.08 pF. It has been observed that the capacitive loading helped in matching the $Z_{in}$ at different resonating bands for different values of reverse bias voltages across the varactor diode.

## 3. Results of Simulation and Measurements

For the proposed meandered loop-slot antenna, the simulation and modeling were performed using HFSS$^{TM}$. The various dimensions of antenna design have been optimized to achieve the sub-1 GHz band. This included optimal varactor diode placement, maximum effective reactive loading, as well as antenna placement on the PCB board. The S-parameters were measured using the Agilent N9918A VNA, while the antenna efficiencies and gain patterns were computed using the SATIMO Star lab anechoic chamber.

### 3.1. Reflection Coefficient Curve

The reflection curves characterize the proposed antenna design. The simulated and measured results of the reflection curves are shown in Figure 5. Figure 5a shows the proposed antenna design simulated reflection coefficient curves ($S_{11}$). The resulted curves are for the design with varactor diode capacitance values from 0.84–5.08 pF and the corresponding reverse bias voltage values ranges from 15 to 0 V. A smooth variation in the resonating bands was observed from 753–1040 MHz with a −10 dB bandwidth of 17 MHz. The wide frequency sweeping with narrowband operations is the proposed antenna design's key characteristic. The proposed slot radiating structure can easily cover several IoT bands in the sub-GHz range.

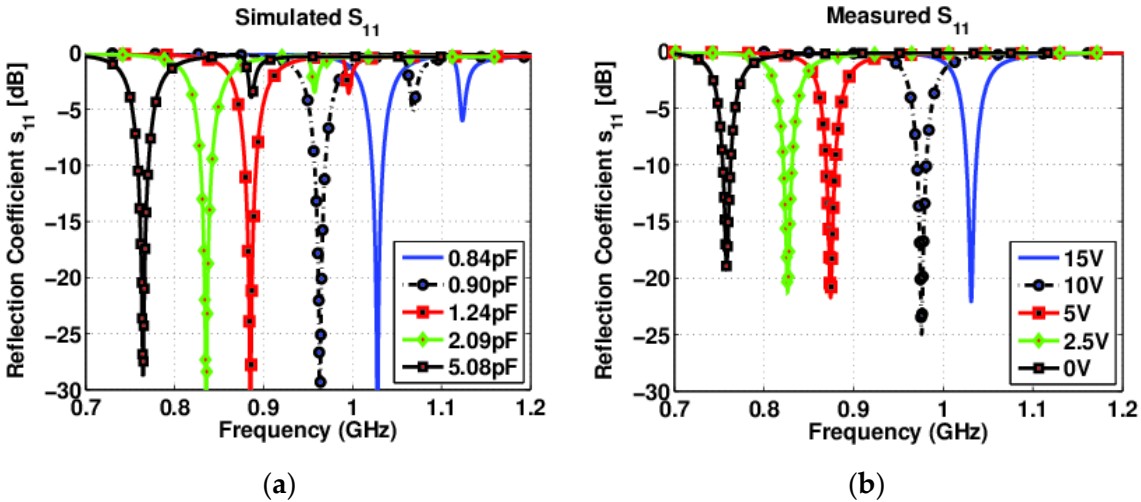

**Figure 5.** Reflection coefficient curves of the antenna (**a**) Simulated $S_{11}$ (**b**) Measured $S_{11}$.

The proposed antenna's measured S-parameters are shown in Figure 5b. It is clear from the figure that the measured results are in good agreement with the simulation results. In the event of any small variation between the two results, it can be compensated for as continuous frequency sweeps have been obtained for both. This demonstrates the main advantage of using varactor diode capacitive loading for these types of operations.

The slight variations in both simulated and measured $S_{11}$ are mainly due to the substrate properties, fabrication tolerances, and varactor diode modeling flexibility in HFSS [44,45]. The HFSS does not consider diode losses and packaging effects. Also, manual soldering of the SMA connector might affect the values of input impedance $Z_{in}$. The additional harmonics are the higher-order resonances of the slot antenna. In an ideal simulating software environment, higher-order modes are poorly matched with the input

impedance $Z_{in}$, and hence are visible. However, the measured $Z_{in}$ is only matched with its fundamental mode of operation within the given frequency bands, and the poorly matched bands did not appear for the measured values. Hence, no harmonics are observed for the measured $S_{11}$ results.

### 3.2. Current Density Analysis

For single antenna designs, current density analysis is usually performed to understand the radiation characteristics, determine the active parts of the radiating structure, and find the effective electrical length of the antenna. For the proposed antenna design, the current density distribution is plotted to understand the antenna's behavior at the resonating frequency bands. It has been observed that current density distribution varies over different operating bands. It has been observed that a large portion of the antenna is radiating at lower frequency bands and vice versa. Such analysis can be utilized to optimize the antenna's dimensions by eliminating non-radiating antenna parts. The surface current distributions of the proposed antenna design were analyzed at various bands of resonance. Figure 6a,b shows the current distribution for non-optimized design for frequency bands at 1.95 GHz and 1.75 GHz, respectively. For the optimized design, the surface current densities at frequency bands of 1029 MHz and 764 MHz are shown in Figure 6c,d, respectively. From the given figures, it can be seen that the current density has a different distribution for different frequency bands. Figure 6d shows the current density distribution at 764 MHz. It can be seen that the maximum current variation is along the outermost part of the loop slot structure. The effective electrical length can be mapped to the corresponding resonating band. The slot length variations affect the first and other resonating bands for different values of capacitive loading. Thus, the analysis helped in understanding the antenna's operation and gave useful insights about operating at different frequency bands.

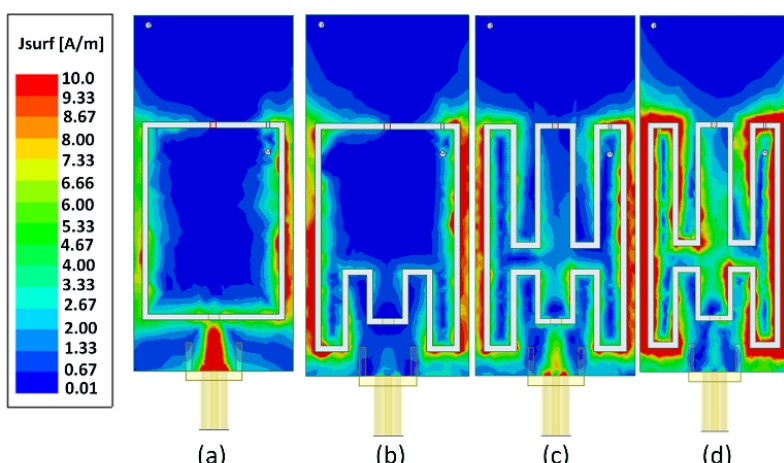

**Figure 6.** Surface current density at (**a**) 1950 MHz, (**b**) 1750 MHz, (**c**) 964 MHz, and (**d**) 764 MHz.

### 3.3. Radiation Patterns

The radiation characteristics of the proposed antenna design at different frequency bands characterize the antenna. The proposed antenna simulated 3-D gain patterns at 885 MHz and 1029 MHz are shown in Figure 7a,b, respectively. The gain patterns of the proposed antenna design, omnidirectional behavior was observed at the resonating bands. The gain patterns that result from simulation and measurements and antenna efficiency (%$\eta$) values of the antenna are listed in Table 1 and are shown in Figure 8. From the resulting values, it can be concluded that the proposed antenna will have good performance when operating in the sub-GHz bands.

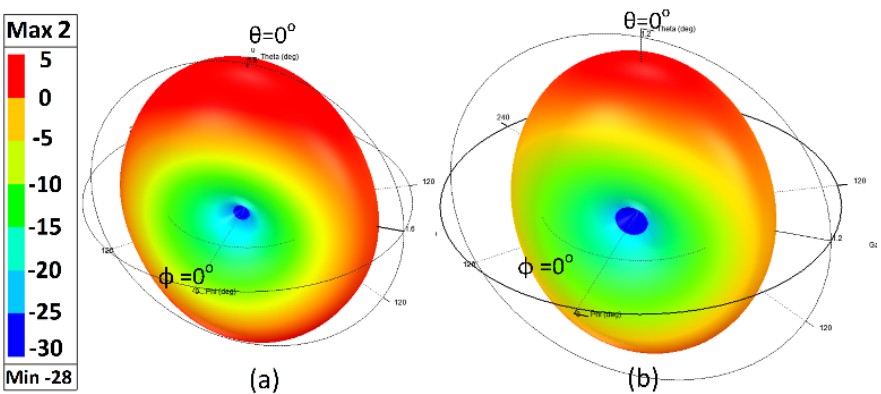

**Figure 7.** Simulated 3-D antenna gain patterns at (**a**) 885 MHz (**b**) 1029 MHz.

**Table 1.** Peak gain (PG) and efficiency (%$\eta$) values of the proposed antenna.

| | Simulated Results | | | Measured Results | |
|---|---|---|---|---|---|
| $f_s$ (MH) | PG (dBi) | %$\eta$ | $f_m$ (MHz) | PG (dBi) | %$\eta$ |
| 764 | 0.86 | 55 | 758 | - | - |
| 836 | 0.98 | 60 | 827 | 0.86 | 54 |
| 885 | 1.2 | 63 | 875 | 0.96 | 59 |
| 964 | 1.4 | 67 | 976 | 1.13 | 64 |
| 1029 | 1.8 | 70 | 1033 | 1.42 | 67 |

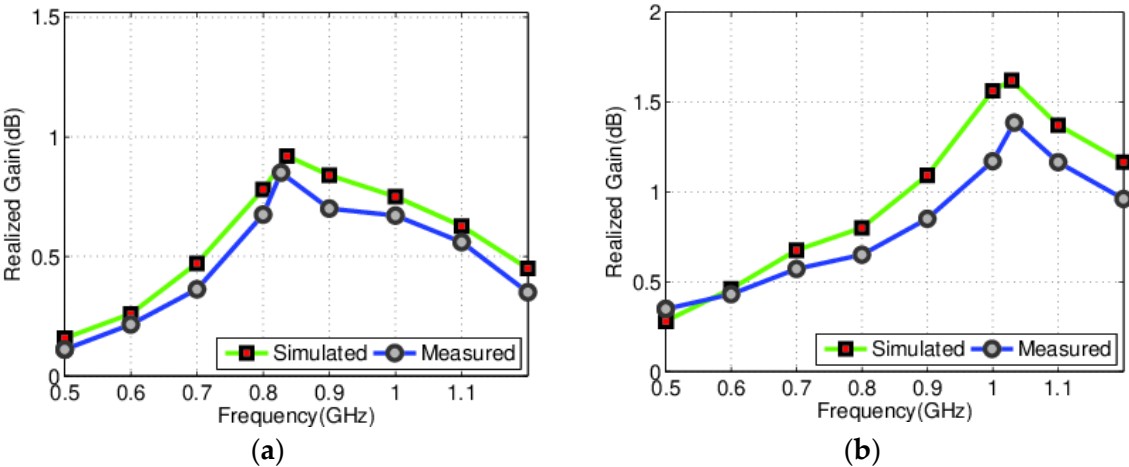

**Figure 8.** Simulated and measured realized gains (**a**) 2.09 pF, 2.5V (**b**) 0.84 pF, 15V.

The proposed antenna design radiation characteristics were validated by performing measurements with the setup shown in Figure 9. The 2D simulated and measured gain patterns at 885 MHz and 1029 MHz are shown in Figure 10 for $\varphi = 0°$ and $\varphi = 90°$ cut. It can be observed that in the given mode of operation at each resonating band, the antenna has omnidirectional behavior. Figure 10a,b show the simulated and measured radiation patterns for $\varphi = 0°$ cut while Figs. Figure 10c,d show the simulated and measured radiation patterns for $\theta = 90°$. The proposed work is intended to be utilized in 5G-enabled IoT devices that mostly operate at lower power. Therefore, with typical signal power densities, this antenna would work well.

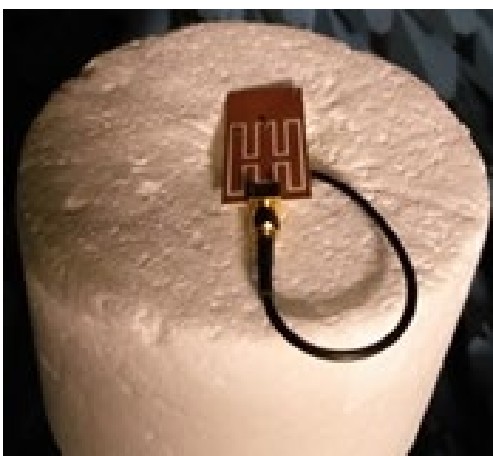

**Figure 9.** Antenna's farfield measurement setup.

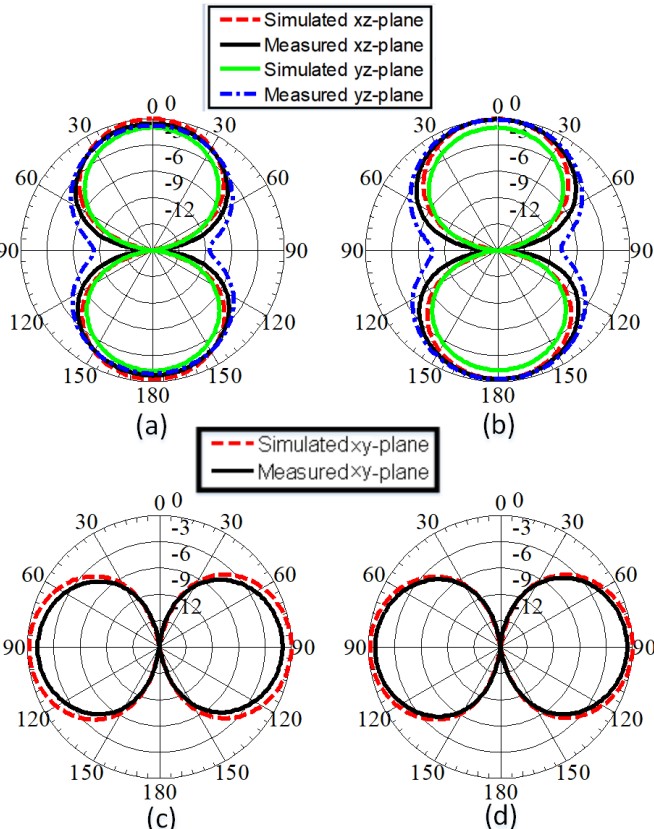

**Figure 10.** Simulated and measured total gain patterns (**a**) φ = 0° at 885 MHz (**b**) φ = 0° at 1029 MHz (**c**) θ = 90° at 885 MHz (**d**) θ = 90° at 1029 MHz.

Table 2 presents a detailed comparison that compares the distinguishing features of the most relevant IoT antennas available in the literature. The various features included antenna type, size, bands covered, suitability for NB-IoT operation, %η, planar structure, percentage miniaturization obtained, continuous frequency sweep, and the number of sub-GHz NB-IoT bands covered. Most of the IoT antennas reported are either wide-band monopole (MP) or non-planar PIFA designs. For such antennas, it is quite challenging to obtain a sub-GHz continuous frequency sweep over a wide band. Although, the proposed antenna design is competitive in terms of its compactness with some of the available designs, it outperformed in sub-GHz bands with wide tunability and NB-IoT operation

for better power management, long battery life, and enabled devices for long-distance communications, etc.

**Table 2.** Proposed IoT antenna versus related works.

| Ref. | Ant. Type | Ant. Size mm²/mm³ | Bands GHz | NB IoT? | %η | Planar? | sub-GHz Bands? | Gain (dBi) |
|---|---|---|---|---|---|---|---|---|
| [2] | monopole | $0.34\lambda_g \times 0.23\lambda_g$ | 2.2, 3.5, 5.5, 8 | No | 54–86.3 | Yes | 0 | 6.9 |
| [3] | monopole | $0.34\lambda_g \times 0.23\lambda_g$ | 3.1–8.5 (wide-band) | No | - | Yes | 0 | −0.78 |
| [4] | monopole | $0.18\lambda_g \times 0.36\lambda_g$ | 0.915, 2.45 | Yes | 93, 95 | Yes | 1 | 1.9, 3.85 |
| [5] | monopole | $0.41\lambda_g \times 0.58\lambda_g$ | 2.4 | Yes | 79 | Yes | 0 | −0.256 |
| [6] | patch | $0.15\lambda_g \times 0.61\lambda_g$ | 0.82 | No | - | No | 1 | −2.5 |
| [15] | monopole | $0.503\lambda_g \times 0.19\lambda_g$ | 0.9, 1.8, 2.3, 2.6 | Yes | 80–90 | Yes | 1 | 3–4 |
| [16] | monopole | $0.15\lambda_g \times 0.16\lambda_g$ | 0.85, 2.1 | No | 40–67.2 | Yes | 1 | 1–3.2 |
| [17] | monopole | $0.14\lambda_g \times 0.1\lambda_g \times 0.28$ | 0.85, 2.4 | No | 70, 90 | Yes | 1 | - |
| [18] | PIFA | $0.23\lambda_g \times 0.05\lambda_g$ | 0.829, 1.95 | No | 60–73 | No | 1 | - |
| [19] | PIFA | $0.044\lambda_g \times 0.16\lambda_g$ | 0.836, 2.2, 3.6 | No | 48–83 | No | 1 | 0.5–5.2 |
| [20] | slot | $0.035\lambda_g \times 0.035\lambda_g$ | 1.32–1.49, 1.75–5.2 | Yes | 45–81 | Yes | 0 | 0.5–4.5 |
| [21] | PIFA | $0.15\lambda_g \times 0.12\lambda_g$ | 0.22, 0.80, 4.7, 4.96 | No | 60–85 | No | 2 | - |
| [22] | patch | $0.75\lambda_g \times 0.75\lambda_g$ | 3.01, 3.83, 4.83, 5.76 | No | 42–60 | Yes | 0 | 1.43–3.06 |
| [23] | PIFA | $1.55\lambda_g \times 1.55\lambda_g$ | 0.41–0.91, 2.1–3.5 | No | 10–50 | No | 1 | −6.1, 4.9 |
| [32] | slot | $1.58\lambda_g \times 1.94\lambda_g$ | 1.82, 1.93, 2.1 | No | 85–89 | No | 0 | 3.2–6.3 |
| [33] | slot | $1.2\lambda_g \times 1.33\lambda_g$ | 5∼6 | No | - | Yes | 0 | 1.36–1.87 |
| [34] | slot | $0.65\lambda_g \times 0.36\lambda_g$ | 3.0 | No | - | Yes | 0 | −0.44–2.97 |
| [35] | slot | $0.55\lambda_g \times 0.16\lambda_g$ | 5.73–5.97 | No | 60 | No | 0 | 1.28 |
| [36] | slot | $0.014\lambda_g \times 0.0425\lambda_g$ | 2.45–3 | No | - | No | 0 | −25 |
| **Prop.** | **slot** | $\mathbf{0.18\lambda_g \times 0.13\lambda_g}$ | **0.758–1.034** | **Yes** | **54–67** | **Yes** | **9** | **0.86–1.8** |

In Table 2, the proposed is compact and better than [2–6,15–23], and [32–36] in terms of antenna size. Although, some work as cited in [4,6,15–19,21,23] covered single sub-GHz bands but without tuning capability to switch between different bands. Some of the designs as presented in [4,6,15–19,21,23] are good for covering the sub-GHz band, but the majority of them are not suitable for NB-IoT operations. Both planar and non-planar IoT antennas, [2–6,15–23,32–36], are reported in Table 2. None of the antennas as reported in [2–6,15–23,32–36] have a continuous frequency sweep in the sub-GHz band. The proposed antenna is the only candidate that can cover sub-GHz bands over a wide frequency range.

## 4. Conclusions

A miniaturized meandered loop slot-line antenna that is suitable for IoT applications was proposed in this work. The presented antenna was optimized using bending, meandering, and reactively loading the slot methods to operate in the sub-GHz bands. Recently, sub-GHz IoT technology is becoming more popular in order to enable devices to achieve long-range communications with low power consumption. A very wideband tuning, 758 to 1034 MHz, was achieved. The antenna's reconfigurability was also investigated in this work. A good understanding of the general antenna design guidelines for this antenna system type was developed. The proposed antenna design was fabricated on a RO4350 substrate with dimensions of 60 mm × 27 mm. The compact planar structure of the antenna, its simple biasing circuitry, its ability to operate over a wide band of sub-GHz, and its narrow-band operation are unique features of this design.

**Author Contributions:** Conceptualization, R.H.; software, R.H., A.M.A. and S.I.A.; validation, R.H. and A.M.A.; writing—original draft preparation, R.H.; writing—review and editing, R.H., N.H., S.I.A. and K.A. All authors have read and agreed to the published version of the manuscript.

**Funding:** This research received no external funding.

**Acknowledgments:** The authors would like to acknowledge the support provided by Researchers Supporting Project number (RSP2022R474), King Saud University, Riyadh, Saudi Arabia.

**Conflicts of Interest:** The authors declare no conflict of interest.

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
