# Peer review of "A Compact Sub-GHz Wide Tunable Antenna Design for IoT Applications"

_electronics, doi:10.3390/electronics11071074_

Round 1

Reviewer 1 Report

“A Compact Narrow-Band Sub-GHz Wide-Band Tunable Antenna Design for IoT Applications” by R. Hussain et al.

 I went through the paper very carefully and thoroughly. Authors studied a compact meandered loop slot-line 5G antenna for Internet of Things (IoT) applications.

1-The paper contains interesting sciences in electronic. The impact of the paper on electronics is going be good. Also, the quality of the research work presented in the paper is also good. .

2-In general, ideas are well explained and understandable but, some tenses, linkers and grammar structures must be checked.

3-The authors should give the thickness and number of layers of all layers that they calculated. Are these parameters obtained from an optimization process?

  1. Authors should obtain the novelty of this manuscript compared to published results?
  2. The authors should argue about the relevance of the temperature dependence of the device.
  3. The Introduction does not provide sufficient background. The introduction does not explain the major contributions and novelty of this work. The significance of the proposed solution has not been summed up.
  4. Authors mentioned “The novelty and distinguishing features of the proposed work are described in detail 107 as below:” 1-8. Authors should explain the evidence to those 8 points comparing the published results.

8- The constructive discussions are missing. As mentioned earlier, authors must make a comparative analysis with other similar solutions and back up their claims on how the proposed solution can be considered as high performing compared to others

9- How their results will be affected if they include energy loss in layers.

10- The novelty of this work should be stated explicitly in the text of the manuscript so that readers can get it easily.

11- Authors should compare their results with the published data and different results.

12- Interpretation of the results is very important, but the authors list the results without sufficient explanation. Authors should be explaining the philosophy of results associates them with the publications and applications.

13- Authors mentioned” The compact planar structure of the antenna, its simple biasing circuitry ability to operate over a wide band of sub-GHz, and its narrow-band operation are unique features of this design” this sentence needs evidences.

14- Authors should be explained the distribution of electric fields with this structure as well as the equations related. It is will be very useful.

15- Authors should explain one or two application to their work.

16- All figures, symbols, equations should be improved.

17- It seems the title need revision by authors to become more informative.

18- The authors should explain the enhancements which added to this study comparing to the published results.

17- How this device can be stable with these kinds of materials.

18- The whole concept is now unclear to a reader what is the actual effect of the sensitivity of this device?

19- Are every term and structure in the proposed design should be clearly and correctly presented not to mislead the reader.

20- Is these measured results or simulated? What kind of simulation/measured is performed? Is authors compared between the both.

21-Finally, I recommend that the paper should be revised taking care of the above comments.

Author Response

Dear Reviewer,

We thank you and really appreciate the respected reviewer for his/her valuable time in reviewing the manuscript and gave us constructive feedback. We have taken into consideration all the comments provided to us in the submitted manuscript. Please find the attached file for point to point response to each comments. The revised manuscript is submitted in addition to this letter. All corrections are marked with Red color in the revised manuscript for ease of tracking.

Thank you 

Reviewer 2 Report

This paper is interesting. The authors present a miniaturized meandered loop slot-line antenna, that is suitable for IoT applications. The proposed antenna was first optimized using bending, meandering, and reactively slot loading methods to operate at the sub-GHz bands, and then fabricated on a RO4350 substrate with dimensions 60 mm × 27 mm, achieving a very wideband tuning 758 - 1034 MHz. The compact planar structure, the simple biasing circuitry ability to operate over a wide band of sub-GHz, and the narrow-band operation are among the advantages of the proposed antenna.

The idea and the concept of the manuscript are attractive for the “Electronics” journal community, and the reported results seem promising. This paper could be useful for researchers in the field, but I strongly believe that its present form needs rectification. Thus I propose “Reconsider after major revision”. The authors are encouraged to submit a revised version of their manuscript, taking into consideration the following major and minor issues:

  1. The manuscript is rather short in comparison to the high quality papers usually published in the “Electronics” journal. The reason for its small size is that it only presents a single prototype, without an actual investigation of at least 3 or 4 similar prototypes, with perturbed sizes, dimensions, positions, substrate and ground plane characteristics, etc. Without investigation, there is the possibility of the bad practice of multiple submissions with slightly different designs with marginal or no novelty.
  2. In Section 1 (Introduction), the novelty of the proposed contribution is not well documented. Related work has been adequately examined, but not in direct comparison with this new one. In order to support the novelty of their work, the authors have to explain why their prototype is novel (i.e. different from those found in bibliography), instead of just stating its advantages. Many antennas have some or all of these advantages, and the readers want to know exactly what this new antenna worth. For example, novel can be the design or the geometry (in this case meandered loop slot-line) of the proposed antenna, when it cannot be found in the present bibliography, while its functional characteristics are fine. Thus, what is the exact novelty (or novelties) of the herein presented antenna, in the context of and in comparison with the previous works? The eight (8) “novelty and distinguishing features of the proposed work” listed are advantages, but not necessary novelties. Table 2 in Section 3.3, which presents a very nice detailed comparison of the distinguishing features of 19 different IoT antennas available in the literature, is successfully giving the context of the authors’ work, but the authors have to highlight the real novelties.
  3. The comparison Table 2 is a strong element of the manuscript, but the relative comments in lines 340-350 are not sufficient. Much more discussion is needed in order to give to the readers a complete overview of the existing technology, emphasizing on the present design.
  4. Equation (1) has to be checked and corrected. The first denominator has to be lm+wm or (lm+wm). Please verify the equation from [35].
  5. Equation (2) has to be checked and corrected. Should tanβLt and tanβ(Lt - Lv) be tan(βLt) and tan(β(Lt - Lv))? Please verify the equation from [36].
  6. There are many spelling and grammatical errors across the manuscript, so the article should be checked throughout and the necessary corrections should be made. For example the verb is missing from the sentence in line 18 “The RO4350 substrate with dimensions 60 mm × 27 mm.”. Another example is the sentence in lines 371-373 “The antenna was optimized using bending, meandering, and reactively loading the slot methods were used to optimize the antenna design to operate at the sub-GHz bands.”, which should be “The presented antenna was optimized using bending, meandering, and reactively loading the slot methods, in order to operate at the sub-GHz bands.”.
  7. The authors have the obligation to thoroughly explain their design procedure. Section 3.3 for antenna design procedure (lines 224-239) is rather short for a “step-by-step” description.
  8. In lines 227-228 the authors state that “The dimensions of the slot were optimized”. Which optimization technique (genetic algorithm, neural network, …) did you use to increase the electrical length of the radiating slot. A short table with several slot dimensions and the corresponding resonation frequencies should be useful for the readers. Similarly in lines 232-233 and 234-235 the authors again state that “The antenna was further optimized by increasing its electrical length” and “The width of each slot as well as the distance between different meandered slots were optimized to tune the antenna”. The optimization technique has to be mentioned and a short table with several slot widths and distances and the corresponding resonation frequencies is needed.
  9. The prototype’s feeding mechanism needs elaboration.

Thus, I propose a major revision by (i) considering and carefully addressing each of the above comments/requests, (ii) exploring, studying and presenting more simulated and fabricated prototypes, (iii) providing and comparing more simulated and measured results, and (iv) emphasizing on the novelty and comparative analysis of the authors’ work, in order for the addressed interesting (but single configuration) problem to be good reference to the “Electronics” researchers and readers..

Author Response

Dear Reviewer,

We thank you and really appreciate the respected reviewer for his/her valuable time in reviewing the manuscript and gave us constructive feedback. We have taken into consideration all the comments provided to us in the submitted manuscript. Please find the point to point response to each comment in the attached file. The revised manuscript is submitted in addition to this letter. All corrections are marked with Red color in the revised manuscript for ease of tracking.
Thank you

Round 2

Reviewer 1 Report

ِAuthors addressed all the comments with detailed analysis.

Author Response

The authors are thankful to the Reviewer for his/her valuable time.